# Too Much of a Good Thing: Rethinking Feed Formulation and Feeding Practices for Zinc in Swine Diets to Achieve One Health and Environmental Sustainability

**DOI:** 10.3390/ani12233374

**Published:** 2022-11-30

**Authors:** Gerald C. Shurson, Pedro E. Urriola, Yuan-Tai Hung

**Affiliations:** 1Department of Animal Science, University of Minnesota, St. Paul, MN 55108, USA; 2Devenish Nutrition, Fairmont, MN 56031, USA

**Keywords:** antimicrobial resistance, bioavailability, biomarkers, environmental impacts, feeding practices, growth promotion, requirements, toxicity, zinc

## Abstract

**Simple Summary:**

Zinc (Zn) is an essential nutrient required by all organisms, but excess use may result in environmental pollution and contribute to the development of antimicrobial resistant bacteria. Although Zn is involved in numerous physiological functions in pigs, accurate determination of dietary requirements has been difficult because homeostasis is tightly regulated, and biomarkers indicative of Zn adequacy are lacking. Dietary Zn concentrations are determined by Zn in common feed ingredients, amount of supplemental Zn added from premixes, use of pharmacological doses of Zn to control post-weaning diarrhea and promote growth, and use of elevated dietary Zn in late gestation to reduce pre-weaning piglet mortality. As a result, Zn concentrations of swine diets are highly variable within countries and across the globe. Dietary Zn regulations in the E.U. were implemented to reduce the negative environmental impacts associated with Zn accumulation in agricultural soils resulting from long-term swine manure application as a consequence of feeding pharmacological Zn concentrations to weaned pigs. However, several alternative feeding strategies could allow the strategic use of elevated Zn concentrations to achieve productivity advantages, while also substantially reducing Zn excretion in manure, which are both important contributors toward improving environmental sustainability of global pork production systems.

**Abstract:**

The objectives of this review were to summarize current knowledge of Zn in swine nutrition, environmental concerns, potential contribution to antimicrobial resistance, and explore the use of alternative feeding strategies to reduce Zn excretion in manure while capturing improvements in productivity. Zinc is a required nutrient for pigs but is commonly supplemented at concentrations that greatly exceed estimated requirements. Feeding pharmacological concentrations of Zn from ZnO to pigs for 1 to 2 weeks post-weaning reduces post-weaning diarrhea and improves growth performance. Feeding elevated dietary levels of Zn to sows during the last 30 days of gestation can reduce the incidence of low-birth-weight pigs and pre-weaning mortality. Most of the dietary Zn consumed by pigs is not retained in the body and is subsequently excreted in manure, which led several countries to impose regulations restricting dietary Zn concentrations to reduce environmental impacts. Although restricting Zn supplementation in swine diets is a reasonable approach for reducing environmental pollution, it does not allow capturing health and productivity benefits from strategic use of elevated dietary Zn concentrations. Therefore, we propose feeding strategies that allow strategic use of high dietary concentrations of Zn while also reducing Zn excretion in manure compared with current feeding practices.

## 1. Introduction

Zinc (Zn) is an essential nutrient and is required in all swine diets for optimal growth and health [1]. However, the Zn requirements of pigs in different stages of growth and reproduction have not been extensively re-evaluated since they were first established in the 1950’s and 60’s, and recommendations vary among various published guidelines [1,2,3,4]. Accurate Zn requirements are difficult to establish because cellular homeostasis is tightly controlled and it is not stored in appreciable amounts in the liver or other body tissues like Cu and Fe, and there are no reliable biomarkers to evaluate Zn status other than growth rate or the development of parakeratosis which may indicate toxicity or deficiency [5,6,7,8]. It is also challenging to accurately quantify Zn concentrations in feeds because of lack of homogeneity of distribution, sampling, and analytical error [9,10], poorly defined bioavailability estimates of organic and nano Zn sources relative to inorganic Zn sources, and potential contamination of feed samples from the environment.

Zinc is also unique among nutrients and most other minerals except Cu, because it provides antimicrobial effects to reduce the incidence and severity of post-weaning diarrhea and promotes growth when added at pharmacological levels (2000 to 3000 mg/kg) from Zn oxide in weaned pig diets [11,12]. However, if diets containing pharmacological concentrations of Zn are fed for more than 3–4 weeks, Zn toxicity may occur [13], except for some Zn sources such as Zn oxide [1]. In addition to using pharmacological Zn concentrations in weaned pig diets, there is emerging evidence that feeding high doses of Zn (up to 600 mg/kg diet) to gestating sows during the last 30 days of gestation improves survivability of low-birth-weight pigs and reducSes pre-weaning mortality of piglets [14]. However, because Zn is not stored in appreciable amounts in the pig’s body, more than 90% of Zn consumed is excreted in manure for subsequent application to cropland [15,16,17]. Studies have shown that swine manure applied to agricultural soils to meet crop removal rates for N and P, results in excess Zn accumulation in soil that can lead to ecotoxicity [17]. Feeding diets containing excess Zn may also contribute to antimicrobial resistance [18]. As a result, the European Union recently imposed regulations [19] that restrict the amount of Zn added to pig diets to 150 mg/kg, and prohibit feeding diets containing pharmacological concentrations of Zn, based on environmental concerns associated with Zn accumulations in agricultural soils resulting from long-term application of manure.

A more holistic approach should be considered when establishing optimal dietary Zn supplementation levels to not only meet Zn requirements, but also achieve post-weaning health and well-being, improve sow productivity, and produce nutritious pork products for human consumption, while avoiding negative environmental consequences and contributions to antimicrobial resistance. This can only be achieved by reviewing current feed formulation and feeding practices that contribute to excess Zn in swine diets, and rethinking approaches for using Zn more strategically and sustainably in pork production systems to achieve One Health and reduce detrimental effects on the environment globally. Therefore, the objectives of this review are to summarize current knowledge of Zn nutrition in swine and estimate the effects of using alternative feeding strategies to limit Zn excretion in manure and environmental impacts.

## 2. Essentiality of Zinc

Zinc is an essential nutrient and the second most abundant trace element in the animal body that provides numerous biological functions [20]. Zinc serves as a cofactor for more than 300 enzymes and 2000 transcription factors [20,21,22,23], is a structural component of bone tissue, and is involved in DNA and RNA metabolism, protein synthesis, gene expression, cell differentiation and proliferation, and cell-mediated immunity [20,24]. However, because Zn is not stored in the body, requirements are met through adequate dietary consumption [5,8,20].

Adequate Zn nutrition plays an important role in maintaining immune functions and reducing inflammation. The anti-inflammatory properties of Zn have been shown in porcine endothelial cells through modulating PPAR-α and -γ to inhibit transcription of NF-κB, which is a key signaling pathway for regulating factors involved in inflammatory responses [25,26]. At weaning, pigs commonly undergo multiple stressors which result in suboptimal feed intake but feeding pharmacological concentrations (2000 to 3000 mg/kg) of dietary Zn from zinc oxide (ZnO) improves innate and adaptive immune capacity of weaned pigs [27]. As a result, elevating dietary Zn concentrations at weaning has been one of the common practices used in swine industry to alleviate intestinal challenges (e.g., enterotoxigenic *E. coli* (ETEC)-induced diarrhea) during nursery phase. Kloubert et al. [27] reported that dietary Zn supplementation at a pharmacological concentration (2500 mg/kg Zn from ZnO) increased phagocytosis and oxidative burst as immune responses in weaned pigs. In addition, weaned pigs fed pharmacological doses of Zn (2500 mg/kg) had a decreased concentration of IL-1β and an increased IL-1RA to IL-1 ratio at 7 days post-weaning, which suggests that Zn alters the inflammatory response and promotes a state of immune tolerance in the intestine [28]. Elevated Zn supplementation (500 ppm) exceeding the Zn requirement has also been shown to increase protein expression of occludin, claudin-1, and ZO-1, while decreasing caspase-9 and -3 activities and inflammatory cytokine expression (i.e., TNF-α, IL-6, IFN-γ) in the small intestine of weaned pigs [29,30], which suggests that Zn plays a role in improving intestinal barrier function. Results from these studies show that Zn supplementation above dietary concentrations required for optimal growth are necessary for optimizing pig health by enhancing immune capacity and overall intestinal function.

Minimizing oxidative stress and inflammation of pigs raised under intensive production conditions is essential for optimizing health and productivity which contribute to achieving One Health and environmental sustainability goals of pork production. Inflammation processes cause the formation of free radicals, and a large amount of free radicals results in oxidative stress [26]. Zinc also plays an important role in reducing oxidative stress of pigs through its functions as a component of endogenous antioxidant defense system. Zinc is a co-factor of superoxide dismutase, which counteracts oxidative stress and inflammation by converting harmful superoxide radicals to oxygen and hydrogen peroxide [26]. In addition, nuclear factor erythroid 2-related factor 2 (Nrf2) is the main transcription factor regulating endogenous antioxidant enzymes and Zn can up-regulate Nrf2 [26]. Weaned pigs fed 500 ppm of nano ZnO had reduced serum malondialdehyde concentrations, increased serum glutathione peroxidase activity, and increased Nrf2 gene expression in the jejunum compared with those fed basal diets [31]. Compared with Zn-deficient pigs, pigs fed 120 ppm of Zn from Zn proteinate increased the activities of superoxide dismutase and glutathione peroxidase in the spleen and liver [32]. Results from these studies show the importance of adequate Zn supplementation to support its role in antioxidant functions and alleviating cell damage induced by oxidative stress.

## 3. Factors Affecting Zinc Digestion, Absorption, Metabolism, Retention, and Excretion

Although current regulations in the European Union require approval of using supplemental trace minerals as feed additive in animal feeds, the bioavailability of trace mineral sources have not been considered [33]. Bioavailability of a trace mineral source is defined as its ability to adequately support the physiological functions of animal metabolism [34], which is more accurate than trace mineral digestion, absorption, or retention that are common misrepresentations [33]. Brugger et al. [33] summarized basic, complementary, and potentially beneficial Zn status measures in farm animals, and they are shown in Table 1.

Zinc absorption and retention is a tightly regulated homeostatic system that results in similar absorption and retention levels regardless of the amount of Zn consumed in feed [5,6,7]. Zinc retention in pigs fed practical diets ranges from 5 to 40% [35,36]. Feeding high dietary concentrations of Zn has been shown to improve pig growth performance [37,38], but about 90–95% of these minerals are excreted [39]. As a result, when high dietary amounts of Zn are consumed above the requirement, fecal excretion increases while absorption and retention is maintained, resulting in lower efficiency of dietary Zn utilization.

Digestion of dietary Zn is influenced by pH in the gastrointestinal tract and the interaction with dietary components, such as phytate, oxalic acid, and polysaccharides that affect the solubility of dietary Zn. Zinc solubility is important for optimizing its use as a nutrient, which is greater in a lower pH in vivo [40] and in vitro [41]. However, phytate, which is naturally present in plant-based feed ingredients, can bind some trace minerals including Zn, Cu, Fe, and Mn, thereby reducing their bioavailability [42,43,44]. Phytate is also dissociated in low pH environment and can form a new phytate complex with divalent cations, such as Zn and Cu in the intestine, thereby limiting the absorption of Zn [45]. In addition to phytate, several diet related factors affect Zn requirements of pigs including Ca, Cu, Cd, Co, histidine, and protein concentrations and sources [1].

The presence of Zn transporter families—SLC30 (ZnT) and SLC39 (ZIP) in the small intestine determines the absorptive capacity of dietary Zn. The transporter ZIP4 plays an important role in Zn uptake from the lumen by increasing cytoplasmic Zn, while ZnT1 and ZnT2 are expressed at the basolateral membrane and mainly function to decrease cytoplasmic Zn [46,47]. Interestingly, the expression of ZIP and ZnT transporters can be influenced by the amount of dietary Zn intake to maintain cellular zinc homeostasis [47,48]. Nursery pigs fed diets with pharmacological levels of Zn (2245 mg/kg) significantly increased expression of ZnT1 and decreased expression of ZIP4 in the jejunum [47]. Divalent metal transporter 1 (DMT1) is also involved in Zn uptake in the enterocytes, but DMT1 is less specific and the expression of DMT1 may be dependent on the Zn source [46], but not the amount of dietary Zn [47]. In addition to the role Zn transporters play in regulating cellular Zn levels, metallothioneins (MTs) serves as a Zn reservoir to regulate intracellular zinc levels and to prevent Zn toxicity [49]. Several of the proteins previously mentioned are essential for maintaining Zn homeostasis and metabolism.

The current dogma of Zn in swine nutrition is that the innate concentrations of Zn in conventional diets is inadequate to meet Zn requirements because of the presence of phytate in grain and oilseed-based feed ingredients due to the formation of insoluble complexes that reduce Zn digestibility [50,51]. Results from several studies have shown that Zn bioavailability and retention is improved when phytase is added to weaned pig diets [52,53]. Bikker et al. [54] conducted a meta-analysis to evaluate the effects of microbial phytase inclusion (150 to 1570 FTU/kg diet) in swine diets and reported that digestibility and plasma concentrations of Zn were improved, and they estimated that 27 mg of Zn from ZnSO_4_ can be replaced with the supplementation of 500 FTU phytase/kg diet. However, these authors acknowledged that this replacement rate may be less in diets containing more than 100 mg Zn/kg diet. Responses may differ based on type of phytase (plant of microbial phytase), diet composition, and amount of heat treatment of the diet. In sows, van Riet et al. [55] suggested that Zn supplementation during gestation may not be needed when feeding diets containing phytase.

**Table 1 animals-12-03374-t001:** Basic, complementary, and potentially beneficial measures of Zn bioavailability in farm animals (adapted from [32]).

Assessment Category	Measure	Notes
Basic		
	Apparent absorption amount of dietary Zn	Total dietary Zn corrected for total fecal losses or apparent absorption as a percentage of dietary intake
	Apparent retention amount of Zn	Total dietary Zn intake corrected for total fecal and urinary Zn losses
	Bone (trabecular bone tissue) Zn	Trabecular bone tissue can be found in the femoral head or tibia
Complementary		
	Serum or plasma Zn	Affected by numerous non-nutritional factors including stress, inflammation, and infection
	Free Zn-binding capacity in serum or plasma	Percentage of free Zn-binding sites in blood plasma as described by [56]
	Liver Zn	-
	Hepatic MT1 gene expression	-
	Milk Zn	-
	Jejunal and colonic relative SLC39A4 gene expression	Some data suggest that the main absorption site for Zn shifts from the small intestine to lower intestinal segments under subclinical deficiency conditions as described by [48]
Potentially beneficial		
	Zn-enzyme activities	Alkaline phosphatase, carbonic anhydrase
	Cell stress signaling pathways for transcription and post-transcription	Regulation of cell stress pathways may provide useful information on the physiological adaptation to Zn deficiency and accumulation in target tissues

## 4. Dietary Zinc Sources

Zinc concentrations in common feed ingredients (Table 2) can provide significant contributions to meeting the pig’s Zn requirement, but they are generally ignored because concentrations vary among sources within ingredients and there is a lack of data on digestibility and bioavailability of Zn in most ingredients [1,57]. Therefore, the amount of Zn provided by major ingredients is destined to be excreted in manure because abundant amounts of Zn are provided to diets at concentrations in excess of the Zn requirements for pigs from trace mineral premixes and supplements.

Although inorganic forms of Zn are relatively inexpensive, they are inefficiently used when included in animal diets because of dietary antagonism [58]. Zinc bound to sulfate rapidly dissociates in solution resulting in the formation of insoluble complexes that reduce intestinal uptake [7,22]. Strong chelating agents can be used to increase trace mineral bioavailability [58], and those with a high affinity to bind trace minerals increase molecular stability after ingestion in the upper gastrointestinal tract, which minimizes formation of insoluble complexes [59,60].

Bioavailability of inorganic Zn sources vary when added to swine diets depending on the types of feed ingredients used [61]. Bioavailability of Zn sources is based on a percentage of a recognized standard and do not reflect the percentage digested, absorbed, and retained [1]. Bioavailability estimates are near 100% for zinc sulfate, zinc carbonate, zinc chloride, and zinc metal dust, but Zn from ZnO is 50–80% available and poorly available from Zn sulfide [60]. Results from several studies suggest that Zn from organic complexes appear to have similar bioavailability to Zn in Zn sulfate [1], but Zn in grains and plant protein sources has low availability due to antagonistic effects of phytate [61].

The use of organic forms of Zn in swine diets have become popular but data comparing their relative bioavailability with inorganic Zn sources is limited. Schelgel et al. [62] conducted a meta-analysis and reported that the bioavailability of organic Zn forms ranged from 85 to 117% compared with inorganic Zn sources and that innate Zn present in common feed ingredients has limited value in pig diets. Hill et al. [63] suggested that organic Zn is utilized differently in weaned pigs than inorganic Zn sources and that the Zn requirement for optimal health of weaned pigs has increased since the NRC (2012) was published.

Zinc oxide nanoparticles are specially manufactured to have a particle size of 1 to 100 nm and are used in many applications including paint, skin lotion pigments, food, electronic appliances, and pharmaceuticals [64]. Although the bioavailability of Zn from ZnO nanoparticles has not been established as a dietary supplement for food producing animals, it is an effective form of Zn for growth promotion due to its antibacterial and immune-modulating effects [64]. However, it is necessary to conduct studies to determine the ecotoxicity of nanoparticles because increased use may compromise soil biodiversity as well as plants, terrestrial and aquatic animals [65].

## 5. Dietary Zinc Requirements, Formulation Methods, Laboratory Analysis

A summary of current recommended zinc requirements of swine in different production phases from various publications is shown in (Table 3). The NRC (2012) recommendation of 26.6 to 46.8 mg Zn per day for piglets weighing 5 to 11 kg have not been revised since 1979. Therefore, these minimum dietary requirement recommendations are no longer relevant and have likely increased due to genetic advances of increased lean growth and increased oxidative stress and immune challenges associated with modern, intensive pig production systems [66,67].

Accurate and repeatable chemical analysis results of nutrients in ingredients and diets is essential for quantifying and assessing adequacy of meeting nutrient requirements of animals. High and inconsistent variation of Zn concentrations in swine diets are commonly observed when subsamples of the same diet are analyzed at different times [9]. Much of this variation can be attributed to sampling procedures used to collect a representative sample. Jones et al. [10] estimated that a minimum of 34 feed samples are needed for analysis to achieve 95% confidence that the analyzed Zn value is within 4 mg/kg of the actual value. However, because collecting this number of subsamples is impractical and costly, accepting results within 15 mg/kg of expected Zn concentration has been suggested as a reasonable accuracy goal, and can likely be achieved by collecting and analyzing 2 to 5 subsamples of a given diet [10].

## 6. Zinc Feeding Practices

### 6.1. Premix Supplementation

Because Zn concentrations in common feed ingredients have generally been considered unavailable and inadequate to meet the Zn requirements of pigs in all phases of production, commercial swine diet are routinely supplemented with Zn from various Zn sources in vitamin-trace mineral premixes. However, because of the challenges of determining Zn requirements in pigs due to the lack of accurate biomarkers, and its relatively low cost compared with other nutrients, Zn is commonly supplemented in swine diets at concentrations that is more than twice that of current recommended requirements. For example, Chae et al. [68] evaluated feeding diets containing 100 to 250% of recommended (NRC, 1998) vitamin and trace minerals concentrations to finishing pigs and observed that feeding the 200 to 250% of requirements resulted in growth performance improvements. Furthermore, as shown in Table 4, recommended dietary concentrations of Zn in each production phase vary between countries and geographic regions.

In the United States, Flohr et al. [69] reported that the average ratio of Zn concentrations of commercial swine diets relative to NRC (2012) recommendations (Table 3) ranged from 30.3 in Phase 1 nursery diets (weaning to 7 kg body weight) to 1.5 in late finishing diets (100 kg to market) and 1.1 for gestation and lactation diets (Table 4). However, some commercial swine operations use diets containing as much as 40 times the NRC (2012) requirement in Phase 1 nursery diets, 2.6 times the requirement in late grower and finisher diets, and 1.7 times the requirement in gestation and lactation diets (Table 4).

In China, Yang et al. [70] reported that the average ratios of dietary Zn concentrations relative to Chinese Feeding Standards [4] shown in Table 3, ranged from 4.9 in Phase 1 nursery diets (8 to 15 kg body weight) to 1.0 in grower diets (25 to 60 kg body weight), and 1.4 to 1.6 in lactation and gestation diets (Table 4). However, maximum Zn supplementation reported by some commercial swine operations in China (Table 4) exceeded those reported in the U.S. survey for grower-finisher pigs (22 times the Chinese Feeding Standards recommendation) and sows (9 to 12 times the Chinese Feeding Standard recommendation).

Similarly, Dalto and da Silva [71] reported average ratios of Zn concentrations in commercial swine diets in Brazil (Table 4) to range from 15.3 in Phase 1 nursery diets (21 to 35 days of age) to 1.3 for grower diets (71 to 120 days of age), with a ratio of 1.0 for gestation and lactation diets relative to Brazil feeding standards [3] shown in Table 3. However, maximum ratios of dietary Zn concentrations relative to Brazilian feeding standards exceeded those reported for the U.S. and China in Phase 2 nursery diets (28), grower diets (36 from 50 to 70 days of age and 3.2 from 71 to 120 days of age), finisher diets (3.6) and gestation (1.8) and lactation (2.0) diets (Table 4). These results indicate that there is significant opportunity to dramatically reduce Zn supplementation concentrations in commercial swine diets in major pork producing countries to meet current feeding recommendations and greatly reduce Zn concentrations in swine manure without prohibiting the strategic use of high dietary Zn concentrations in early nursery and late gestation diets to enhance productivity.

**Table 4 animals-12-03374-t004:** Summary of dietary Zn supplementation concentrations in major pork producing countries.

Country	Weighted Mean	Mean	Minimum	Median	Maximum
United States [69]					
Nursery—Phase 1 (weaning to 7 kg BW)	3173	3032	1906	2931	4002
Nursery—Phase 2 (7 to 11 kg BW)	2340	2081	75	2050	3294
Nursery—Phase 3 (11 to 23 kg BW)	673	401	66	120	3030
Grower (23 to 55 kg BW)	86	99	30	110	150
Grower (55 to 100 kg BW)	78	85	30	89	131
Finisher (100 kg to market)	72	74	30	75	131
Gilt development (20 kg to breeding)	105	122	61	124	174
Gestation	113	123	57	125	165
Lactation	113	123	57	125	165
Boars	122	143	84	130	279
Brazil [71]					
Nursing piglets (3 to 20 days of age)	-	2225	120	2480	3488
Nursery—Phase 1 (21 to 35 days of age)	-	1876	90	2388	3488
Nursery—Phase 2 (36 to 49 days of age)	-	996	89	150	3075
Grower (50 to 70 days of age)	-	448	67	100	3200
Grower (71 to 120 days of age)	-	103	44	100	250
Finisher (121 days of age to market)	-	90	10	90	235
Gilt development	-	119	88	120	170
Gestation	-	111	12	103	195
Sow transition diets	-	121	100	120	144
Lactation	-	110	12	103	220
Boars	-	112	12	112	195
China [70]					
Nursing piglets (birth to 8 kg BW)	-	425	23	100	1980
Nursery—Phase 1 (8 to 15 kg BW)	-	534	0.12	133	1980
Nursery—Phase 2 (15 to 25 kg BW)	-	95	0.12	77	1980
Grower (25 to 60 kg BW)	-	61	0.12	58	275
Grower (60 to 90 kg BW)	-	62	0.12	59	1075
Finisher (90 kg to market)	-	63	0.12	60	1075
Gilt development	-	69	30	70	123
Gestation	-	72	0.12	70	530
Lactation	-	72	0.12	70	460
Boars	-	69	30	64	220

### 6.2. Premix Withdrawal

Numerous studies (Table 5) have been conducted to evaluate growth performance, carcass characteristics, and tissue concentrations of trace minerals resulting from partially or completely removing trace mineral premixes without or with vitamins from growing-finishing pig diets for varying lengths of time before slaughter [72,73,74,75,76,77,78,79,80,81,82,83]. Most studies showed that 100% withdrawal of the vitamin-trace mineral premix up to 6 weeks before market had no effect on growth performance and carcass traits [72,73,74,76,78,80,81]. For example, Mavromichalis et al. [73] conducted 3 experiments to determine the effects of removing vitamin and trace mineral premixes during the last 30 days of the finishing period before slaughter and showed no differences in growth performance, carcass characteristics, and muscle quality but reduced cost and excretion of excess nutrients. However, although results from one study showed that 100% removal of vitamin-trace mineral premix from the entire grower-finisher phase had no effect on growth performance and pork quality [80], fasting plasma glucose concentrations, carcass backfat, and liver weight as a percentage of body weight increased, while ham weight and Zn concentrations in muscle, liver, and bone decreased when no trace mineral premix was added. Results from other studies indicated that withdrawal 12 weeks prior to market tended to decrease growth performance [77], and 100% removal for the last 4 weeks before market reduced growth rate and feed conversion [76]. None of the studies evaluating growth performance from feeding diets containing 50% of recommended vitamin-trace mineral concentrations for up to 9 weeks before market showed detrimental effects [76,78,81,83]. In addition, most studies evaluating reduced Zn supplementation levels showed no effects on enzyme activity [79], blood hemoglobin and hematocrit [81], and pork quality [79,80], but increasing the length time (0 to 6 weeks) of supplemental Zn (100% of NRC requirement) withdrawal from the diet linearly decreased liver and metacarpal bone Zn concentrations [82]. However, it is important to recognize that studies reporting negative effects of reduced inclusion of premixes include all vitamins and trace minerals and not only Zn. Therefore, it appears that a feasible strategy to reduce Zn excretion in swine manure would be to reduce Zn supplementation to 50% of recommended levels during the last 30 days before slaughter without compromising growth performance, health, or pork quality.

### 6.3. Pharmacological Levels Post-Weaning

An initial study conducted by Poulsen [84] showed that feeding a weaned pig diet containing 3000 mg/kg Zn from ZnO for 14 days postweaning reduced postweaning diarrhea and increased growth rate. Since this initial discovery, numerous studies have been conducted to confirm these responses when feeding pharmacological concentrations of Zn (i.e., 2000 to 6000 mg/kg) for up five weeks to weaned pigs, but some studies have shown improvements in growth rate without reducing postweaning diarrhea [1]. However, growth performance responses were inconsistent from a meta-analysis evaluating results from 26 studies that evaluated the dietary addition of pharmacological concentrations of Zn from ZnO for pigs post-weaning [11]. In addition, reductions in the incidence of diarrhea have been inconsistently observed among studies that evaluated feeding the combination of pharmacological concentrations of Zn and Cu to weaned pigs, and additive effects may or may not occur [1]. Schweer et al. [85] evaluated growth performance responses from feeding various alternatives to feeding sub-therapeutic antibiotics as growth promoters from about 2000 feeding trials and reported that improvements in growth performance were observed on average in only 30% of the trials. However, feeding growth promoting concentrations of zinc and/or copper resulted in more consistent improvements in growth rate (38.7% of trials), average daily feed intake (24% of trials), and gain efficiency (Gain:Feed; 19.4% of trials).

Although ZnO has been the most commonly used source of Zn used at pharmacological concentrations in diets, a limited number of studies have shown that other forms of Zn, such as Zn methionine, can also be used to provide similar benefits at lower doses [86,87,88]. However, five other sources of supplemental organic Zn were not effective in promoting growth of weaned pigs when added to diets at 500 mg/kg compared with growth improvements from adding 500 or 2000 to 2500 mg/kg Zn from ZnO [89]. The likelihood of Zn toxicity from feeding pharmacological levels of Zn depends on Zn source, dietary concentration, duration of feeding, and concentrations of other interacting minerals in the diet [1]. Zinc oxide nano particles appear to be an effective alternative to pharmacological use of conventional ZnO because it has better antimicrobial properties than conventional ZnO [90] and can be used at lower doses to reduce environmental contamination [64,91].

A recent study conducted by Hansen et al. [92] evaluated increasing dietary levels of Zn from ZnO in weaned pig diets to determine the optimal dietary levels of Zn and showed a quadratic feed intake and growth response during the first two weeks after weaning. These researchers indicated that 1400 mg Zn/kg diet and 400 mg Zn consumed/day were optimal, and to avoid an increased risk of diarrhea, a diet containing 1100 mg Zn/kg or consumption of 166 mg Zn/day during the first week post-weaning was required. As a result, it appears that the Zn requirement for newly weaned pigs is much greater than the current EU regulation of a maximum dietary Zn concentration of 150 mg/kg diet. Therefore, in order for a 7 kg pig to meet the recommended Zn intake of 48.6 mg/day, it would need to consume 312 g feed/day, which is highly unlikely because feed consumption is usually less than 235 g/day [93,94,95,96].

Oh et al. [97] conducted a study to compare a ZnO-glycine chelate and nano particle-sized ZnO with the standard form of ZnO on diarrhea score, nutrient digestibility, Zn utilization, intestinal immune responses, and fecal microflora of weaned pigs and showed that feeding diets containing 200 mg/kg Zn from ZnO nano particles provided similar positive effects compared with feeding 2500 mg/kg Zn from ZnO. Nano particles of ZnO has been shown to have antimicrobial activity against *E. coli* and improve intestinal morphology similar to feeding high doses of standard ZnO [31,98].

### 6.4. Elevated Dietary Zn Levels in Late Gestation

Hedges et al. [99] demonstrated that feeding a corn-soybean meal gestation diet containing 33 mg/kg Zn through five parities supported optimal gestation performance but was inadequate during the lactation period for sows. However, feeding diets containing only 13 mg/kg Zn during the last four weeks of gestation prolongs the farrowing process [100]. Feeding Zn deficient diets to gilts during gestation and lactation has been shown to reduce litter size and birth weights of piglets as well as serum and tissue Zn concentrations [101,102,103,104,105].

Zn deficiency is associated with intrauterine growth retardation, decreased birth weight, suboptimal immune and neurological system development, and increased preweaning mortality in rats and humans [106,107]. Feeding diets containing high Zn concentrations to sows reduced the incidence of stillborn pigs [104] and increased litter birth weight [108]. Zinc accumulates in high concentrations in the conceptus [109] and after day 90 of gestation, maternal liver Zn concentrations decrease, and fetal liver Zn concentrations increase [110]. Vallet et al. [111] showed that elevated dietary Zn concentrations during late gestation reduced preweaning mortality of low-birth-weight pigs. This response was confirmed in a study by Holen et al. [14] where gestating sows were fed diets containing one of three dietary Zn concentrations (125, 365, and 595 mg/kg Zn) from Zn sulfate and a Zn amino acid complex for the last 30 days of gestation under commercial production conditions. Piglets from sows fed 365 mg/kg Zn had heavier birth weight and tended to have heavier birth weights from sows fed the diet containing 595 mg/kg Zn compared with piglets from sows fed the low Zn diet. As a result, mortality of low-birth-weight pigs was less for sows fed the highest Zn diet and overall mortality tended to decrease as dietary Zn concentration increased. These results indicate that feeding elevated dietary Zn to sows during the last 30 days of gestation increases survival of low-birth-weight pigs and reduced overall mortality. This improvement in productivity reduces the environmental impact of pork production by producing more viable pigs per sow using existing feeding levels.

### 6.5. Comparison of Zn Consumption per Market Hog Produced among Different Feeding Scenarios

Determining optimal dietary concentrations of Zn requires a more holistic approach than previously considered in developing the current regulations in the European Union. Therefore, it is worthwhile to consider various feeding scenarios that allow strategic use of high dietary concentrations of Zn for relatively short time periods (weaning to 21 days post-weaning and the last 30 days of gestation) to gain productivity improvements when daily feed intake is relatively low, while minimizing or removing Zn supplementation in diets in all other stages of production. As previously described, there is some evidence that innate Zn concentrations in common feed ingredients used in swine diets may provide adequate Zn nutrition if phytase is added to diets to release and utilize a significant portion of natural Zn [53,54,55,56]. Several studies have also shown that reducing excessive Zn supplementation in premixes can dramatically reduce Zn excretion in manure without compromising pig health and performance [83,112,113,114]. Substantial evidence also supports the complete [72,73,74,75,77,79,81] or partial [76,78,81,83] removal of vitamin-trace mineral premixes during the last 4 to 6 weeks of the finishing period to reduce excess Zn and other trace mineral excretion in manure without detrimental effects on growth performance and carcass composition. Furthermore, except for pharmacological doses of Zn in weaned pig diets, the current maximum allowable Zn concentrations in growing-finishing pig and sow diets in the European Union (Table 3) are greater than those recommended in the United States [1], China [4], and Brazil [3]. Therefore, it is useful to compare different dietary Zn feeding strategies and the estimated impact of the amount of Zn consumed per market hog produced, to determine the relative impact on Zn excretion in manure (Table 6).

Trace mineral concentrations in common feed ingredients are typically not considered when formulating swine diets. Using Zn concentrations of common feed ingredients from NRC [1], Kerr et al. [57], and INRAE-CIRAD-AFZ [115] to formulate a basal diet comprised of corn, soybean meal, wheat, and wheat middlings, and DDGS without a vitamin-trace mineral premix provides about 32, 50, 57, and 41 mg/kg of dietary Zn in the nursery, grow-finish, gestation, and lactation diets, respectively. If Zn present in these ingredients were 100% available to the pig, which it is not, it would contribute to about 32%, 80%, 57%, and 41% of NRC [1] recommended dietary Zn concentrations for nursery, grower-finisher, gestation, and lactation diets, respectively. Because total dietary Zn concentrations in swine diets are under scrutiny, naturally occurring Zn in raw materials should be considered when formulating diets.

As previously described, there are substantial differences in recommended requirements or feeding guidelines for Zn in various pig production phases (Table 3). However, because feed consumption and recommended dietary Zn concentrations vary among production phase, it is necessary to consider the relative contribution of Zn consumed in each production phase to produce a market hog to calculate total Zn consumption, which is directly proportional to Zn excretion in manure. Using dietary Zn requirement estimates [1,2,3,4] with the assumptions described in the footnote in Table 6, China has the lowest Zn consumption per market hog produced followed by the United States, Brazil and European Union (Scenario 1 in Table 6). Because the recommended maximum dietary Zn concentrations at various stages of production are greater for the current European Union guidelines compared with China and the United States (Table 3), the estimated Zn consumption per market hog produced when using increased Zn supplementation levels for nursery pigs and gestating sows (Scenario 4 and 5) was comparable to that for the European Union (55,124 mg Zn/pig in Scenario 1) which does not allow comparable high Zn doses in these production phases. When comparing average dietary Zn supplementation from industry surveys in the United States [69], China [70], and Brazil [71] (Scenario 6 and 7) to increased Zn supplementation levels for nursery pigs and gestating sows in Scenario 4, the Zn consumed per pig marketed was 28% greater in the United States and 59% greater in Brazil but was 29% less in China. These results indicate that except for China, current average feeding levels of Zn in the United States and Brazil are unnecessarily high compared with the proposed optimal feeding scenario 4. Further reductions in Zn consumed per market hog produced could be realized if Zn supplementation from vitamin-premixes was removed during the last 30 days before market without adversely affecting growth performance or carcass composition as shown for Scenario 8. Using the feeding practice of Zn withdrawal from premixes during the last 30 days of the finishing period, estimated Zn consumption per market hog produced can be reduced by 9745 mg Zn/pig in the United States, by 6064 mg Zn/pig in China, and by 6449 mg Zn/pig in Brazil. Lastly, except for the European Union, the lack of any restrictions on maximum Zn supplementation in swine diets has resulted in some commercial swine operations feeding Zn far beyond acceptable standards. In the scenario of using maximum Zn supplementation levels in these countries (Scenario 9 and 10), an excess of 91,910 to 252,266 mg Zn per market hog produced occurs on some swine farms in the United States, China, and Brazil compared with using elevated Zn concentrations in early nursery and late gestation diets and following current recommended Zn requirements (Scenario 4) in these countries.

There are other possible scenarios that can also be considered. For example, Hanson et al. [92] showed that nursery pigs should consume 166 mg Zn daily (diets containing 1100 mg Zn/kg) during the first week post-weaning to reduce the risk of post-weaning diarrhea and require a daily intake of 400 mg Zn for the optimal growth response in the first two weeks post-weaning resulting in a total Zn intake of 5600 mg per pig. Therefore, pharmacological Zn supplementation in the nursery phases in Europe would be feasible if removing Zn supplementation in the last grow-finish phase that could create space for about 6000 to 9000 mg of Zn intake per pig to maintain current Zn intake and excretion levels. In addition, the combined use of pharmacological concentrations of Zn in nursery diets and elevated Zn in late gestation diets is feasible if average Zn supplementation levels in all countries were reduced to simply meeting current recommended levels. As shown in Table 6, the difference between current average dietary Zn supplementation levels and requirements or guidelines for the United States (34,969 mg Zn/pig) and Brazil (65,542 mg Zn/pig) provides ample space to use the combined elevated dietary Zn strategies (about 6000 mg Zn/pig in nursery and 3000 mg Zn/pig = 9000 mg Zn/pig) and provide an overall reduction in Zn use per pig marketed in these countries. However, for China, the use of the combined elevated dietary Zn feeding strategy would increase Zn consumption per market hog by 16,631 mg Zn/pig compared with current average Zn supplementation levels from the industry survey (Scenario 7) but would reduce Zn consumption by 252,266 mg Zn/pig on farms using maximum dietary Zn supplementation rates (Scenario 10). Therefore, there are several feasible Zn feeding strategies that can be implemented globally that allow strategic use of elevated dietary Zn to improve swine health and productivity while reducing Zn excretion to the environment compared with current feeding practices. 

## 7. Effects of Dietary Zn Concentrations and Sources on Pork Quality

Several studies have evaluated the effects of dietary supplementation of various Zn sources and concentrations on growth performance and carcass characteristics of growing-finishing pigs [116,117,118,119,120], which showed inconsistent responses to feeding dietary Zn concentrations greater than 50 to 60 mg/kg from 25 to 135 kg of body weight recommended by NRC [1]. Gowanlock et al. [81] indicated that there is an adequate amount of innate microminerals (Cu, Fe, Mn, and Zn) in typical corn-soybean meal diets to meet the dietary requirements for growth, and there were no detrimental effects of eliminating these trace minerals from diets including no effects on loin pH, color, and drip loss. Limited studies have been conducted to determine Zn effects on pork quality, but of those, the focus has generally focused only on chemical composition, drip loss, pH, color, and fatty acid concentrations [121,122] and not shelf-life stability which may benefit from dietary Zn supplementation by Zn functioning in the promoting antioxidant defense system [123]. Although feeding two dietary concentrations of supplemental Zn from Zn glycinate to growing-finishing pigs had no effects on trace mineral concentrations, color stability, and fatty acid profile of meat, nor was lipid oxidation in raw meat and meat homogenates incubated with pro-oxidant catalysts affected, supplementing diets with 45 mg/kg Zn from Zn glycinate was effective in limiting lipid oxidation in cooked meat. Therefore, there may be some benefit to improve oxidative stability of pork by supplementing grower-finisher diets to meet the current NRC [1] Zn requirements.

## 8. Environmental Loading of Zinc from Various Feeding Practices and Comparison with Sewage Sludge

The application of pig manure to agricultural land has many beneficial effects on soil properties including increasing organic carbon, and available phosphorus and potassium concentrations in surface soil [124]. In addition, pH, electrical conductivity, total carbon and nitrogen concentrations were greater in soil amended with pig manure applications during a 13-year time period compared with soils amended with commercial fertilizers [125]. As a result, the application of pig manure to agricultural land reduces the environmental footprint of pig production by recovering and recycling nutrients back to soil for crop production which minimizes the need for fossil fuel derived fertilizers. Although the application of animal manure to agricultural soils as a nutrient source for crop production is generally considered to be an environmentally friendly practice, there are also environmental risks associated with surface and ground water pollution, and soil microbial ecosystems if excessive amounts of potentially toxic elements are applied [126,127]. Leclerc and Laurent [128] summarized inventories of the release of 8 metals from the application of manure to agricultural soils in 215 countries from 2000–2014 and observed that the toxic impacts per area of agricultural land were greater in the EU compared with Southeast Asia, and Hg, Cu, and Zn were the major contributors to global impacts. Xu et al. [129] evaluated the long-term accumulation of Cu and Zn in soil and plants from using different pig manure application rates over a 10-year time period and reported that soil concentrations increased by 204% and 107%, respectively, at the high application rate of 15.73 kg Zn/ha/year. This resulted in some leaching in tested soil, but the Cu and Zn concentrations measured were below concentrations considered phytotoxic to crops. Similarly, Wu et al. [130] reported that long-term (17 years) application of swine manure to soil increased soil Zn concentration by 18.9-fold compared with the control.

In many countries, high concentrations of Zn and Cu in livestock manure limits its application to agricultural soils due to excessive accumulation of these elements [112,131,132,133]. Feeding pharmacological doses of Zn from ZnO to weaned pigs has become an environmental concern because increased concentrations of Zn pig diets is directly proportional to increased Zn concentrations in pig manure, and the subsequent manure application rate is directly proportional to soil Zn concentration [124]. As a result, several studies have shown that reducing the amount of dietary Cu and Zn in swine diets is beneficial for the environment [112,113,114]. Feeding high dietary concentrations of Cu and Zn to pigs reduces natural microbial activity and organic matter fermentation in anaerobic deep pits used for manure storage which results in manure removal and handling problems. When swine manure is applied to cropland, the relatively high application rate is usually based on an amount necessary to meet the nitrogen or phosphorus needs of crops but greatly exceeds the rate of Zn uptake by crops [16,17]. As a result, Zn accumulates in soil over several years of application, which can contribute to run-off of surface water, and ground water contamination can occur [134], causing ecotoxicity which alters ecosystems, biodiversity, and environmental sustainability of food production and the natural environment. These negative environmental impacts led the European Union to ban the use of pharmacological levels of Zn in June 2022 [12] and establish regulations [19] limiting maximum Zn concentrations in all swine diets, with the goal of reducing excess Zn in swine feeds, manure, soil, and the environment. While this initially appears to be a prudent approach for improving the nutritional efficiency of Zn and reducing its negative impacts on the environment, it prohibits the opportunity to achieve gains in productivity which positively affect sustainability of pork production.

In north and south China, animal manure was identified as the greatest contributor of Cu and Zn to agricultural soils compared with sewage sludge, sewage irrigation, inorganic fertilizers, and atmospheric deposition [135]. However, Alengebawy [136] indicated that inorganic phosphorus fertilizer can contain 50 to 1450 mg/kg Zn, while nitrogen fertilizers (1 to 42 mg/kg Zn), lime fertilizers (10 to 450 mg/kg Zn), and livestock manure (15 to 250 mg/kg Zn) contain much lower concentrations of Zn worldwide. Smith [137] summarized metal concentrations in various types of compost and sewage sludge applied to agricultural soils and showed than Zn and Pb were found in the greatest concentrations, with maximum Zn concentration of 2330 mg/kg in mechanically segregated compost. Latosińska et al. [138] summarized current permissible concentrations of metals in municipal sewage sludge for natural use and reported values (dry matter basis) of 500–1000 mg/kg in China, 2500 mg/kg Zn in Poland, 2500–4000 mg/kg Zn in the European Union, and 2800 mg/kg Zn in the United States and South Africa. Zinc has been identified as the main element in sewage sludge treated agricultural soils that is of environmental concern because it is relatively labile, is readily absorbed by plant tissues compared with Cu, and adversely affects soil microbial activity [137]. Previous studies have reported that 60% of sewage sludge is applied to agricultural soils in France and 57% is applied to crop land in Belgium [139]. The median worldwide soil Zn content was 90 mg/kg with maximum limits of 400 mg/kg Zn based on the voluntary PAS100 standard in the U.K. and 2800 mg/kg Zn established by the U.S. Environmental Protection Agency [137].

Zhang et al. [140] determined the optimal application rate of pig manure containing 1115 mg/kg Cu dry mass and 1497 mg/kg Zn dry mass to maintain fertility and soil quality and recommended an application rate of 50 to 100 tonnes/ha as an environmentally sustainable practice, which is a greater rate than recommended by Lee et al. [124]. Ding et al. [141] conducted a study to determine mutually beneficial doses of dietary Cu and Zn that promote growth and health while minimizing excretion in manure of pigs from 9 to 60 kg body weight and suggested 50 mg/kg Zn from 9 to 25 kg BW and 10/mg/kg Zn from 25 to 60 kg BW.

However, the most extensive evaluation of the impact of different Zn feeding scenarios was conducted by Dourmad and Jondreville [16] using assumptions from manure Zn concentrations, excretion, application to crop land, crop Zn uptake from a previous study [15], and results are summarized in Table 7. The 250 mg/kg Zn feeding level was the maximum allowed in the E.U. prior to 2003, when it was reduced to a maximum of 150 mg Zn/kg diet [16]. Feeding a phase 1 nursery diet containing 2500 mg/kg Zn to weaned pigs from 8 to 13 kg body weight resulted in 15.8 g/pig greater intake and excretion of Zn in manure than feeding 250 mg/kg Zn diets throughout the production cycle (Table 7). However, the 250 mg/kg Zn feeding level is greater than the median Zn supplementation levels for U.S. Phase 3 nursery diets, Brazil Phase 2 nursery diets, and China Phase 1 and 2 nursery diets as well as grower-finisher and sow diets from all 3 countries (Table 4). In fact, the 250 mg/kg Zn supplementation level is even greater than maximum Zn levels for grower-finisher pig and sow diets reported in the U.S. and for grower-finisher diets (71 days of age to market) and sow diets in Brazil (Table 4). The scenario of feeding 150 mg/kg Zn in all diets is similar to the current EU regulations [16] except that the maximum diet concentration of Zn in the grower-finisher phase is 120 mg/kg, which was 150 mg/kg as in the calculations shown in Table 7. The scenario of meeting Zn requirements in each phase is the most conservative Zn feeding scenario because currently recommended Zn requirements for weaned pigs (70 to 150 mg/kg), growing-finishing pigs (50 to 120 mg/kg), and sows (45 to 150 mg/kg) summarized in Table 3 for the U.S., Brazil, China, and E.U. generally meet or exceed the Zn concentrations in Table 7, except for the China recommendation of 45 mg/kg Zn for gestating sow diets. Dourmad and Jondreville [16] suggested that the amount of supplemental Zn can be reduced when adding phytase to swine diets because results from a previous study [142] showed that the addition of 500 phytase units/kg of diet was equivalent to supplying 30 mg/kg Zn as Zn sulfate. These comparisons show that strategic use of pharmacological concentrations of Zn from ZnO to promote growth and reduce post-weaning diarrhea while prudent use of Zn supplementation (i.e., less than 150 mg/kg Zn) could reduce Zn excretion by 52% to as much as 92% if reducing Zn supplementation levels to the requirement levels shown in Table 7.

## 9. Zinc Contributions to Ecotoxicity

Among various heavy metals and trace elements of environmental concern, Cd, Pb, and Hg are generally considered to be highly toxic to humans and animals, but less toxic to plants, while Cu, Zn, and Ni are more toxic to plants than to animals and humans [138]. Zinc is naturally found in soils at concentrations of 10 to 300 mg/kg [143], with origins from parent rock erosion, atmospheric deposition from volcanic ash, forest fires, and dust [144]. Human induced Zn introductions into the environment include combustion of fossil fuels, galvanization, tire and railing rust, motor oil, cement, tar production, and hydraulic fluid [145] as well as animal feeds.

Excessive Zn accumulation in soil and water supplies is toxic to plant and animal life because Zn is non-degradable in the environment [12,146]. Soil Zn concentrations greater than 90 mg/kg can adversely affect plants, humans, and microbial ecosystems [147]. The World Health Organization [148] established maximum permissible limits of Zn concentrations in soil (50 mg/kg) and plants (0.6 mg/kg) which are less than for Pb (85 and 2 mg/kg, respectively) but much greater than Cd (0.8 and 0.2 mg/kg, respectively), with allowable Cu concentrations of 36 and 10 mg/kg, respectively. However, current guidelines for maximum Zn concentrations in agricultural soil range from 120 to 250 mg/kg in the U.S. and Canada [149], with higher levels allowed for residential soil (250 mg/kg Zn) and industrial and commercial soils (410 mg/kg Zn).

Toxic concentrations of Zn in agricultural soils are a significant global concern because they cause reduced plant growth due to reduced photosynthesis and enzyme activity as well as having adverse effects on the soil microbiome [150]. Zinc toxicity in soil results in changes in bicarbonate and organic matter content, and soil pH [136]. Zinc is toxic at elevated concentrations to bacteria because it competes with other metals binding to active sites of enzymes [151] and interferes with enzyme activity and cell performance in soil and plants [152]. Zinc toxicity in plants results in oxidative stress, Fe deficiency, growth inhibition, membrane degradation, chlorophyll degradation, altered mineral nutrition, decreased seed germination, disruption of enzyme activities and under extreme toxicity, plant death [149]. Excessive Zn in plant cells causes disruption of physiological processes including prevention of element transport from the root system to leaves, leading to eventual plant death [136]. Acute Zn intoxication in humans is a relatively rare event that only occurs from long-term, high dose Zn supplementation because it is relatively harmless compared with other metal ions with similar chemical properties [153]. However, humans are less sensitive to Zn concentrations in water than aquatic organisms [154].

Although the accumulation of Zn in agricultural soils is a hazard with the potential to cause toxicity to soil microorganisms, plants, and other living organisms, the actual risk of deleterious effects of Zn is dependent on the likelihood of exposure and the magnitude of the consequence (risk = likelihood of exposure × magnitude of consequence). The likelihood of Zn accumulation in soil is dependent on numerous factors including amount of Zn in manure, manure application rate, soil texture, soil hydrology, vertical transport of Zn to lower layers of soil, crop Zn removal rates, among others. Likewise, the magnitude of impact of accumulated Zn in the soil is dependent on many factors such as the presence and distribution of microorganisms, insects, and plants that are dependent on the soil and their sensitivity to Zn. Consequently, the quantification of risk requires the use of modeling approaches that include the likelihood of exposure and the magnitude of the effect. The European Food Safety Authority conducted a risk assessment using predicted no-effect concentration (PNEC) models to estimate risk of surpassing PNEC of Zn in various types of soils, which ranged from 120 to 340 mg Zn/kg [17]. Modeling soil Zn accumulation for about 100 years, Monteiro et al. calculated that PNEC would not be reached in 6 of 10 soil modeling scenarios [17]. However, in 4 out of 10 scenarios, soil Zn concentrations would surpass PNE concentrations in about 100 years. These results indicate that soil Zn accumulation varies significantly by soil type and sustainable management of Zn requires identifying areas where Zn application to soil should be restricted and areas suitable for increased Zn application.

There is also increasing interest in using bioremediation practices to remove toxic trace elements effectively and efficiently from contaminated soil and wastewater [146,150,155]. Studies have shown that inoculating Zn contaminated soils with certain strains of Zn tolerant bacteria can be effective bioremediation approach for alleviating phytotoxic effects of Zn on corn production [150]. Phytoremediation is one cost-effective mitigation strategy that could effectively revegetate soils containing high concentrations of heavy metals [156].

Livestock and poultry are relatively tolerant of consuming high amounts of dietary Zn, but responses vary by species and the relative concentrations of calcium, copper, iron, and cadmium, which can reduce the toxic effects of Zn when fed at elevated concentrations in diets [21]. Source of Zn also affects the likelihood of toxicity due to differences in relative bioavailability. For example, studies evaluating the effects of feeding 2000 to 4000 mg/kg Zn from ZnO showed no signs of Zn toxicity, but pigs fed diets containing 1000 mg/kg Zn from Zn lactate became lame and unthrifty after 2 months [1]. The first signs of Zn toxicity in all species include depressed feed intake and growth [21]. Other signs of Zn toxicity observed in pigs include lethargy, arthritis, hemorrhage in axillary spaces, gastritis, and death in growing pigs, as well as reduced litter size and piglet weight at weaning and osteochondrosis when feeding toxic dietary concentrations of Zn to sows [1]. Therefore, the toxicity of Zn in animal diets depends on Zn source, concentration, duration of feeding, concentrations of phytate and other minerals in the diet, and animal species.

## 10. Zinc Contributions to Antimicrobial Resistance

Antimicrobial resistance is a major global threat to human health and was associated with 4.95 million deaths in 2019 [157]. Excessive use and misuse of antibiotics in human medicine and animal agriculture has led to selection for antimicrobial resistant bacteria through the spread of resistance genes to pathogens, especially *Staphylococcus* spp. and *E. coli* [158,159]. Subtherapeutic use of antibiotics as growth promoters in swine diets has been banned in the European Union and is restricted through the Veterinary Feed Directive in the United States, has led to the use of a wide variety of alternative feed additives as growth promoters in weaned pig diets. As a result, use of Cu and Zn at pharmacological concentrations in weaned pig diets have become preferred alternatives because of their high efficacy compared with other types of feed additives [85,134]. Pharmacological doses have been widely used to effectively control post-weaning diarrhea and E. coli F4 infections in weaned pigs which contribute to significant economic losses in commercial pork production systems [12]. However, exposure to Zn may contribute to antibiotic resistance even without the presence of antibiotics [158].

Although bacterial resistance to metals has been occurring long before resistance to antibiotics was discovered, very little is known about the epidemiology because there are no methodologies to determine susceptibility testing or thresholds established for phenotypic testing which make it difficult to study [160]. Therefore, most studies are based on the determining the presence of known resistance genes. Several studies have shown a genetic linkage through co-resistance of metal and antibiotic resistant genes, where genes involved in resistance to metals are often observed in plasmids next to antimicrobial resistant genes in bacteria isolated from humans, domestic animals, and food products [160]. Additional studies have provided evidence that pharmacological doses of Zn in diets for weaned pigs is a potential contributor to co-selection antimicrobial resistance [158,161,162,163,164,165,166,167]. Indirect co-selection of antibiotic resistance may also be possible through chromosomal metal resistance genes [168], and co-selection of antimicrobials and metals can occur through shared mechanisms of resistance (i.e., cross-resistance), altered expression of antimicrobial resistance genes after metal exposure (co-regulation), or from biofilm formation [160]. Therefore, the co-occurrence of Cu and Zn and antibiotic resistance genes in animal isolates of multi-drug resistant *Salmonella* and methicillin-resistant *Staphylococcus aureus* (MRSA) demonstrates a direct genetic linkage of resistance genes indicating that Cu and Zn can potentially cause antibiotic resistance development of human pathogens [18]. However, data are lacking for whether Zn-resistant bacteria can acquire antibiotic resistant genes and become resistant to antibiotics and if antibiotic resistant bacteria are more capable of being Zn-resistant than bacteria that are susceptible to antibiotics [158].

Development of antimicrobial resistance is dependent on several complex ecological processes that involve acquisition, resistance gene transfer to naïve microorganisms, persistence, and retention of resistance despite no antimicrobial use associated with co-resistance, cross-resistance, and co-regulation. Co-resistance, cross-resistance, and co-regulation processes can occur over a wide range of antibiotic and metal concentrations [169]. The minimum selective concentration, which is a threshold concentration that resistance genes will be selected, has been determined for several antibiotics, but these were developed without considering the potential effect of co-resistance [170]. No clear increases have been observed in soil bacteria antimicrobial resistance after long-term application of sewage sludge on farm soils [171]. In addition, although elevated dietary concentrations of Cu are also a concern for acquisition and persistence of antimicrobial resistance, no Cu induced co-selection of antimicrobial resistance has been observed in pigs fed 250 mg/kg Cu vs. 20 mg/kg during a 116-day feeding experiment [172]. Therefore, there is a lack of information to support that decreased use of pharmacological doses of Zn and Cu may decrease the prevalence of antimicrobial resistance in feces, manure, and soil in pig farms.

## 11. Conclusions

We propose that alternative approaches be considered to balance the improved health and productivity benefits from the strategic use of elevated dietary Zn in weaned pig and late gestation diets, with minimizing the detrimental effects of Zn accumulation in soil, effects on ecotoxicity, and potential contributions to antimicrobial resistance. These approaches include: (1) developing a database of ingredients and Zn sources to formulate swine diets on a digestible Zn basis, (2) restricting dietary digestible Zn concentrations to a narrower margin in excess of the requirement, (3) using alternative Zn sources to achieve growth and reproductive responses at Zn concentrations below the current pharmacological levels of Zn from other conventional inorganic Zn sources being used, (4) developing prescriptive strategic pharmacological Zn supplementation protocols that allow the dynamic use of elevated Zn for limited periods of time (i.e., first 2 weeks post-weaning and last 30 days of gestation), while restricting dietary Zn concentrations to meet requirements during the remainder of the production phases, and (5) withdrawal of Zn from vitamin-trace mineral premixes during the last 4 to 6 weeks of the finishing period before market.

## Figures and Tables

**Table 2 animals-12-03374-t002:** Zinc concentrations (mg/kg dry matter) in common feed ingredients used in swine diets [1,57].

Ingredient, mg/kg DM	[1]	[57] ICP ^1^
Corn	19	23
Wheat	35	39
Barley	31	37
Sorghum	17	31
Soybean meal, dehulled, solvent extracted	54	61
Soy protein concentrate	ND	30
Canola meal, solvent extracted	54	ND
Corn DDGS	58	114
Wheat midds	103	115
Meat and bone meal	101	112
Spray dried plasma	14	17
Blood meal	53	31
Fish meal	95	120
Limestone	ND	18
Dicalcium phosphate	ND	95
Monocalcium phosphate	ND	105
Sodium chloride	ND	36
ZnO	72%	798,837
ZnSO_4_	35.5%	382,343
CuSO_4_	ND	1145

^1^ ICP = Inductively coupled plasma-mass spectroscopy analysis.

**Table 3 animals-12-03374-t003:** Summary of dietary Zn requirements and recommendations for swine in various stages of production from various publications.

Country	Recommended Dietary Zn Concentration, mg/kg
United States [1]	
Nursery—Phase 1 (5 to 7 kg BW)	100
Nursery—Phase 2 (7 to 11 kg BW)	100
Nursery—Phase 3 (11 to 25 kg BW)	80
Grower (25 to 50 kg BW)	60
Grower (50 to 75 kg BW)	50
Finisher (75 to 100 kg BW)	50
Finisher (100 to 135 kg BW)	50
Gilt development (20 kg to breeding)	-
Gestation	100
Lactation	100
Boars	50
Brazil [3]	
Nursing piglets (3 to 20 days of age)	-
Nursery—Phase 1 (4 to 15 kg BW)	123
Nursery—Phase 2 (15 to 30 kg BW)	110
Grower (30 to 50 kg BW)	88
Grower (50 to 70 kg BW)	77
Finisher (70–100 kg BW)	66
Finisher (100 to 120 kg BW)	55
Gestation ^1^	110
Lactation ^1^	110
Boars ^1^	110
China [4]	
Nursing piglets (birth to 8 kg BW)	110
Nursery—Phase 1 (8 to 15 kg BW)	110
Nursery—Phase 2 (15 to 25 kg BW)	70
Grower (25 to 60 kg BW)	60
Grower (60 to 90 kg BW)	50
Finisher (90 kg to market)	50
Gilt development	70
Gestation	45
Lactation	50
Boars	75
European Union [2] ^2^	
Weaning to 12 weeks post-weaning	150
Growing pigs (25 to 135 kg BW)	120
Gestation	150
Lactation	150

^1^ Only one recommendation is given for all breeding swine [3]. ^2^ Only maximum allowed total Zn concentrations provided based on European Union regulations.

**Table 5 animals-12-03374-t005:** Summary of studies evaluating the effects of vitamin trace mineral (VTM) premix withdrawal on growth performance, carcass characteristics, meat quality, tissue concentrations, and other physiological measures.

Reference	Production Phase (Initial BW)	Added Dietary Zn, mg/kg to the Basal Diet	Withdrawal Period	Key Findings
[72]	Finishing pigs (80 kg)	60	100% withdrawal for 3 or 5 wk prior to market	Removing VTM premix had no negative effect on growth performance and carcass yield, lean, and fat
[73]	Finishing pigs (86 kg)	110 (based on NRC 1998 recommendations)	100% withdrawal for 4 wk prior to market	Removing VTM premix had no negative effect on growth performance, gastric morphology, and carcass traits (i.e., yield, longissimus muscle area, longissimus muscle color traits, cooked meat tenderness)
[74]	Finishing pigs (94 kg)—Exp. 1 Finishing pigs (79 kg)—Exp. 2	150	100% withdrawal for 30 d	Removing VTM premixes had no negative effect on growth performance, mortality, and behavioral aberrations (i.e., tail biting, ear biting)
[75]	Finishing pigs (91 kg)	Not reported in VTM premix	100% withdrawal prior to market	Removing the VTM had no negative effect on growth performance, within-pen variability in final weight, carcass characteristics (i.e., yield, backfat thickness, fat-free lean), and integrity of vertebrae
[76]	Finishing pigs (85 kg)	0–100 (0% to 200% of NRC 1998 concentrations)	50% withdrawal (Exp. 1) or 100% withdrawal (Exp. 2) for 4 wk	Reducing VTM premix by 50% had no negative effects on growth performance while removing 100% of VTM significantly decreased ADG and FCR; 50% or 100% VTM premix withdrawal may decrease pork shelf-life (i.e., increased thiobarbituric acid reactive substances and peroxide value)
[77]	Growing pigs (54 kg)	142	100% withdrawal for 6 or 12 wk prior to market	Removing VTM for 12 wk had a tendency to decrease poor growth performance but no effects on carcass yield, fat-free lean, backfat depth, loin muscle area, and pH
[78]	Growing pigs (21 kg) Finishing pigs (54 kg)	30–120 (50% or 200% of NRC 1998 recommendations) 22.5–90 (50, 100, or 200% of NRC 1998 recommendations)	50% withdrawal for 6 wk in growing phase 50% withdrawal for 9 wk in finishing phase	Reducing 50% of VTM level suggested by NRC (1998) had no negative effects on growth performance and meat quality (i.e., loin muscle area, pH, drip loss, cooking loss, shear force, tenderness, and flavor) of growing pigs and finishing pigs
[79]	Weaning to finishing pigs	100 (based on NRC 1998 recommendations)	100% withdrawal for 4 wk prior to market	-Removing VTM premixes had no negative effect on growth performance, carcass traits (i.e., yield, loin muscle area, backfat), longissimus dorsi Zn, Cu/Zn superoxide dismutase or glutathione peroxidase activity
[80]	Growing to finishing pigs (22–109 kg)	154	100% withdrawal in grow-finish phase	Removing VTM premixes had no negative effects on growth performance and pork quality (i.e., color, marbling, shear force, cooking loss)
[81]	Growing to finishing pigs (24–115 kg)	0–50 (0, 50, 100% of NRC 1998 recommended concentrations of Cu, Fe, Mn, and Zn)	50 or 100% withdrawal for 3 growth phases	Reducing VTM premix by 50% or 100% had no effects on growth performance, blood hemoglobin and hematocrit, and carcass traits (i.e., longissimus muscle area, backfat, loin pH, loin color, and drip loss)
[82]	Weaning to finishing pigs (7–83 kg)	0–100 (0 or 100% of NRC 1998 concentrations of Zn from inorganic or organic sources)	100% withdrawal for 0, 2, 4, and 6 wk (80–120 kg BW)	Increased length of deletion period linearly decreased liver Zn and metacarpal bone Zn concentrations; Organic sources had greater deposition in visceral organs than inorganic sources
[83]	Finishing pigs (85 kg)	82.5–165 (Met or exceeded NRC 2012 recommendations)	50% withdrawal during the last two dietary phases prior to market	Reducing VTM premix by 50% had no effect on growth performance, carcass traits (i.e., hot carcass weight, backfat, loin depth, and lean percentage), and serum Zn and Cu concentrations but significantly decreased Zn and Cu excretion in feces

**Table 6 animals-12-03374-t006:** Estimates of Zn consumption (mg) per market hog produced under various feeding scenarios in major pork producing countries ^1^.

Feeding Scenario	U.S.	China	Brazil	E.U.
(0) Estimated innate Zn consumption from feed ingredients	15,433	15,590	15,366	15,433
(1) Meet current Zn requirements or guidelines with no safety margin, mg/pig	34,359	33,424	39,961	55,124
(2) As for (1) + feed 2000 mg/kg in Phase 1 and 2 nursery diets	51,197 (+49%)	54,725 (+64%)	63,303 (+58%)	-
(3) As for (1) + feed increased Zn to gestating sows during the last trimester before farrowing	37,535 (+9%)	36,953 (+11%)	43,073 (+8%)	-
(4) Combined increases in Zn supplementation in nursery and gestating sows	54,373 (+58%)	58,254 (+74%)	66,415 (+66%)	-
(5) Relative differences of (4) compared with EU regulations in (1)	−751 (−1.4%)	+3130 (+6%)	+11,291 (20%)	-
(6) Zn consumption based on average Zn supplementation levels from industry surveys	69,328	41,623	105,503	-
(7) Relative differences of (6) compared with (4)	14,955 (+28%)	−16,631 (−29%)	39,088 (+59%)	-
(8) As for (6) if VTM premix withdrawn in last phase before market	59,583 (−14%)	35,559 (−15%)	99,054 (−6%)	-
(9) Zn consumption based on maximum Zn supplementation levels from industry surveys	146,283	310,520	309,385	-
(10) Relative differences of (9) compared with (4)	91,910 (+169%)	252,266 (+433%)	242,970 (+365%)	-

^1^ The following assumptions were used to calculate estimated Zn consumption (mg) per market hog produced under different feeding scenarios: gestation = 115 days, lactation = 21 days, daily gestation feed intake = 2.2 kg, daily lactation feed intake = 6 kg were constant across four countries/regions. The estimated feed intake and estimated days on feed within a growth phase were calculated based on a computer model of NRC 2012 [1] for the scenario in the United States, China, and the European Union. Brazilian tables for poultry and swine [3] were used to estimate feed intake and days on feed for the scenario in Brazil.

**Table 7 animals-12-03374-t007:** Estimated Zn concentration in manure from pigs under different feeding scenarios (adapted from [14,15]).

	Zn Feeding Scenario			
Measure	Pharmacological Use of Zn Post-Weaning Followed by 250 mg/kg in Remaining Diets	250 mg/kg in All Diets	150 mg/kg Zn in All Diets	Meet Zn Requirements
Diet Zn concentration, mg/kg				
Phase 1 nursery (8 to 13 kg BW)	2500	250	150	70
Phase 2 nursery (13 to 28 kg BW)	250	250	150	50
Growing-finishing (28 to 110 kg BW)	250	250	150	30
Sows	250	250	150	70
Zn balance (0 to 110 kg BW), g/pig				
Intake ^1^	84.1	68.3	41.7	9.0
Excretion ^2^	81.7	65.9	39.3	6.7
Manure Zn, mg/kg DM ^3^	2542	2128	1269	284
Years to reach 150 mg Zn/kg soil DM ^4^	79	95	167	1160

^1^ Based on 1.4 kg feed/kg gain for Phase 1 nursery, 1.9 kg feed/kg gain for phase 2 nursery, and 2.9 kg feed/kg gain during the growing-finishing phase. ^2^ Body Zn concentration calculated as Zn (mg/pig) = 21.8 × empty body weight (kg). ^3^ Assumed 460 L of manure produced per pig, including 50 L water, and containing 7% dry matter. ^4^ Assumed annual manure application based on 170 nitrogen/ha, 3000 tonnes soil/ha, annual uptake of Zn by crops of 200 g Zn/ha, and initial soil concentration of 20 mg/kg DM.

## Data Availability

All data provided in this manuscript were adapted from other published sources and were appropriately cited in the tables and reference section.

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
