# Peer review of "Too Much of a Good Thing: Rethinking Feed Formulation and Feeding Practices for Zinc in Swine Diets to Achieve One Health and Environmental Sustainability"

_animals, 2022, doi:10.3390/ani12233374_

Round 1

Reviewer 1 Report

I found the paper really interesting.

Lines 616-618: Please, delete. It has been repeated in lines 612-613

Author Response

We agree with the reviewer's comment and have deleted this sentence.

Reviewer 2 Report

The review  summarize current knowledge of Zn in swine nutrition, environmental concerns, potential contribution to antimicrobial resistance, and explore the use of alternative feeding strategies to reduce Zn excretion. The Authors have analyzed the topic in the light of the bibliography. They've read and interpreted the studies and summarized the results expressing their point of view.

I think the topic is relevant for professional who deal with the swine species who can make use of this publication to be aware of the state of the art on the subject.

The review brings together numerous (172) and appropriate references carried out on the subject.

It is a review and in their conclusions they summarize  what can be done to improve.

The conclusions are consistent with the evidence and arguments presented and they address the main issues posed.

The work is well organized  and abundant and appropriate references have been considered.

please check the References in text 

The manuscript can be accepted in the present form.

Author Response

Reviewer 2

The review summarize current knowledge of Zn in swine nutrition, environmental concerns, potential contribution to antimicrobial resistance, and explore the use of alternative feeding strategies to reduce Zn excretion. The Authors have analyzed the topic in the light of the bibliography. They've read and interpreted the studies and summarized the results expressing their point of view.

I think the topic is relevant for professional who deal with the swine species who can make use of this publication to be aware of the state of the art on the subject.

Response: Thank you.

The review brings together numerous (172) and appropriate references carried out on the subject.

Response: Thank you, we agree.

It is a review and in their conclusions they summarize what can be done to improve.

The conclusions are consistent with the evidence and arguments presented and they address the main issues posed.

Response: Thank you.

The work is well organized and abundant and appropriate references have been considered.

Please check the References in text 

Response: The references in the text have been checked

The manuscript can be accepted in the present form.

Response: Thank you.

Reviewer 3 Report

Comments to authors

Title: Too much of a good thing: Rethinking feed formulation and feeding practices for zinc in swine diets to achieve One Health and environmental sustainability”.

Manuscript ID: animals-2062798.The review is covering may be all studied related to Zn in pig diets and authors have make their great efforts for writing this information in the present manuscript. I recommended Minor Revision. I have some comments should be addressed.

General comments:

1-     Simple Summary: should be more concise

2-     Tables should be followed the journal guidelines (only table 5 is the correct to follow the guidelines)

3-     Line 10 please use Zinc (Zn) as the first showing the abbreviation.

4-     Line 47 I think it should be [1-4] not [1,2,3,4]. Revise it in all the manuscript.

5-     Line 52, authors should mention the sources of Zn such metallic, organic, or nano sources.

6-     Line 78-81 this sentence is not understood, needed to rewrite

7-     Line 118-199 add citation

8-     Line 124-126, authors just mentioned that Zn decreased MDA, what about the other indices such Protein carbonyl, MYO, 8-OHdG (DNA damages) please insert some data about that if are available.

9-     Line 210 but Zn in grains and plant protein sources has low availability [60]. I think due to its highly amounts of P.

10- Line 220, I think there are two types of zinc nanoparticles synthesis by green methods (Bacteria and phytochemical) or chemical method and the source or zinc used in this manufacturing such Zinc oxide is the best one for Zinc nanoparticles.

11- Line 306, removing vitamins correct it.

12- Zinc contributions to ecotoxicity

In this section, authors did not demonstrate any studies regarding the toxicity effects of Zn or higher levels of Zn in the diets of animals. Only they mentioned mostly the studies related to soil and plants, I know animals fed on those plants contained high levels of Zn.

Please revise this part carefully and add some data if available.

13- Conclusions

Needed further future perspectives

Author Response

Reviewer 3

Title: Too much of a good thing: Rethinking feed formulation and feeding practices for zinc in swine diets to achieve One Health and environmental sustainability”.

Manuscript ID: animals-2062798.The review is covering may be all studied related to Zn in pig diets and authors have make their great efforts for writing this information in the present manuscript. I recommended Minor Revision. I have some comments should be addressed.

General comments:

1-     Simple Summary: should be more concise

Response: Our simple summary fits within the word limits for this section of the journal and is difficult to make any more concise than it already is due to the breadth of topics discussed. No change was made.

2-     Tables should be followed the journal guidelines (only table 5 is the correct to follow the guidelines)

Response: We do not understand this comment. All table were originally formatted the same, and there are no differences in format between table 5, which was identified as correct, and the other tables. We assume that the final format will be modified by the editors so that the information in columns and rows is in alignment with information presented in the original version submitted. No change was made.

3-     Line 10 please use Zinc (Zn) as the first showing the abbreviation.

Response: Defined as suggested on line 10 and again on line 44 for the introduction.

4-     Line 47 I think it should be [1-4] not [1,2,3,4]. Revise it in all the manuscript.

Response: This was corrected throughout the manuscript.

5-     Line 52, authors should mention the sources of Zn such metallic, organic, or nano sources.

Response: Sentence has been revised to include Zn sources.

6-     Line 78-81 this sentence is not understood, needed to rewrite

Response: Sentence has been revised.

7-     Line 118-199 add citation

Response: Citation has been added.

8-     Line 124-126, authors just mentioned that Zn decreased MDA, what about the other indices such Protein carbonyl, MYO, 8-OHdG (DNA damages) please insert some data about that if are available.

Response: While there are studies showing the detrimental effect of Zn deficiency on DNA damage in rats and human models, we were unable to find studies evaluating the effects of feeding nano ZnO or organic Zn sources on these indices in pigs, which is the context of this discussion.

9-     Line 210 but Zn in grains and plant protein sources has low availability [60]. I think due to its highly amounts of P.

Response: We revised to indicate that low Zn availability is primarily due to antagonistic effects of phytate.

10- Line 220, I think there are two types of zinc nanoparticles synthesis by green methods (Bacteria and phytochemical) or chemical method and the source or zinc used in this manufacturing such Zinc oxide is the best one for Zinc nanoparticles.

Response: Yes, there are different types of Zn nanoparticles.

11- Line 306, removing vitamins correct it.

Response: No change. The studies cited and discussed withdrew vitamin-trace mineral premixes, not just trace minerals or Zn, from the diet for varying lengths of time before market.

12- Zinc contributions to ecotoxicity

In this section, authors did not demonstrate any studies regarding the toxicity effects of Zn or higher levels of Zn in the diets of animals. Only they mentioned mostly the studies related to soil and plants, I know animals fed on those plants contained high levels of Zn.

Please revise this part carefully and add some data if available.

Response: New text has been added on lines 714-727.

13- Conclusions

Needed further future perspectives

Response: We revised this section with recommendations of alternative approaches to consider to address this challenge.
